# Nonceliac Gluten Sensitivity—A Masquerading IBS or a Real Phenomenon?

**Yoram Elitsur *** and **Deborah Preston**

Department of Pediatrics, Gastroenterology Division, Marshall University School of Medicine, Huntington, WV 25701, USA; preston28@marshall.edu

* Correspondence: elitsur@marshall.edu

**Abstract:** The gluten-free diet has become popular among the public. People who are using this diet have reported symptom relief once gluten has been removed from their diet. Nonceliac gluten sensitivity (NCGS) has emerged as a new diagnosis for those patients who have tested negative for celiac disease. Although there are no diagnostic markers established for NCGS, its symptomatology ranges from gastrointestinal symptoms to neuropsychiatric symptoms. Indeed, some of these symptoms are also seen in patients with irritable bowel syndrome (IBS), such as abdominal pain, bloating, altered bowel movement, diarrhea, and constipation. It is important to add that unlike celiac disease, NGCS has never been associated with any long-term malignancy. We aim to review the recent clinical data available on this topic and address the overlap symptoms between NCGS and IBS. We concluded that despite the overlap symptoms between both diseases, NCGS is a real clinical phenomenon that awaits its own diagnostic clinical criteria and specific laboratory markers. We suggest that patients with gluten sensitivity who are negative for celiac disease should be considered for NCGS.

**Keywords:** gluten; NCGS; IBS; celiac disease

## 1. Introduction

Nonceliac gluten (wheat) sensitivity (NCGS) has emerged as a new clinical entity with no specific diagnostic tools. In Western societies, avoiding gluten has become a social phenomenon supported by public magazines and social media. The gluten-free market in the U.S. was estimated at $2.5 billion in 2010 [1]. It is estimated that gluten-free diets have been practiced by about 7–15% of adults in different countries [2]. Unfortunately, when examined scientifically by double-blind placebo control studies, only a small fraction of those self-declared gluten sensitivities were found to be correct [3]. In the present review, the clinical diagnosis, symptoms, and association with IBS will be discussed.

## 2. Epidemiology

Nonceliac gluten sensitivity (NCGS) is a clinical entity that defines gastrointestinal or extraintestinal symptoms related to consumption of gluten or wheat proteins. Currently, the exact gluten-related protein that is responsible for NCGS pathophysiology has not been clearly identified, and as such, wheat protein has been included in its pathophysiology. Accordingly, the diagnosis of NCGS could only be done after celiac disease and wheat allergy were ruled out [4]. Today, there are no specific markers to make the diagnosis, and most of the patients are self-diagnosed. Indeed, review of the published data around the world has shown that the global rate of self-proclaimed NCGS in adults ranged between 6.2% in the Netherlands and 14.9% in Australia [2]. Few studies in children reported rates between 4% and 12% [1,4]. In another report, the northern latitude of the United States had a higher rate of NCGS in adults, and those with higher income, higher education, less BMI, and non-Hispanic ethnicity were

at high risk for the disease [5]. Due to lack of accepted clinical tests or biomarkers for the disease, the rates of missed, or over, diagnosis of NCGS were high. In order to improve diagnostic yield, some investigators used double-blind placebo control (DBPC) methods. In those reports, the missed diagnosis was reported in up to 60% of the self-declared NCGS [6], and the Salerno Expert Criteria reported that most of the studies found gluten challenges to significantly increase symptom scores compared with placebo, and 16% showed gluten-specific symptoms [3]. The data suggest that even with DBPC studies, the diagnostic accuracy of NCGS is not optimal.

## 3. Symptoms

Nonceliac gluten sensitivity (NCGS) has wide-ranging symptoms, including gastrointestinal and extraintestinal symptoms. Studies showed that many of the gastrointestinal symptoms were similar to the symptoms reported in patients with irritable bowel syndrome (IBS), such as abdominal pain, bloating, altered bowel movement, diarrhea, and constipation. The nongastrointestinal symptoms included symptoms in the neuropsychiatry field, such as headache, foggy mind, tiredness, anxiety, not feeling well, and others [7,8]. Indeed, the symptoms reported by NCGS patients scatter between IBS-like symptoms and psychogenic-like symptoms such as functional abdominal pain or functional dyspepsia. Some investigators placed NCGS symptoms between IBS and celiac disease symptoms [9].

## 4. Pathophysiology and Treatment

The pathophysiology of NCGS has yet to be determined. Several optional triggers have been identified, including gluten, amylase/trypsin inhibitors, and others. In a DBPC study, Di Sabatino A et al. showed that gluten, in patients with self-reported NCGS, was the major trigger for the gastrointestinal and nongastrointestinal symptoms [10]. Others showed that with enzymatic destruction of gluten, clinical symptoms were improved [11]. Alpha-amylase/trypsin inhibitors (ATI) are part of the wheat protein molecule, and were suggested as a culprit in the symptoms of patients with celiac disease and NCGS. In a recent study, investigators have examined the ATI contents in ancient wheat species (spelt, emmer, and einkorn) that were hypothesized to be less toxic due to the possible low bioactivity of ATI. The authors found that except Einkorn, all ancient wheat species contain high levels of ATI, and thus may trigger symptoms in NCGS patients [12]. Others found that like in other inflammatory processes, the ATI and/or gluten proteins activate the innate immune system by increasing IL-8 levels and other inflammatory cytokines [13,14].

In other studies, fructans were shown to trigger symptoms in patients who were self-diagnosed with NCGS [15]. In addition, further improvement of symptoms in NCGS patients was noted when a GFD and low fructan diet (low Fermentable Oligosaccharide, Disaccharide, Monosaccharide, and Polysaccharide [FOD-MAP] diet) were used compared to those who were on a GFD alone [16]. Others reported that low FOD-MAP diets may improve symptoms in self-reported NCGS patients [17].

## 5. NCGS and Celiac Disease

NCGS and celiac disease have gluten and gluten-related proteins (ATI) as the major culprits [10,13,14,18]. In addition, in both diseases intestinal microbiota and some mucosal immune alteration (innate immune response) are also involved [18,19]. Nonetheless, in spite of these similarities, the diseases are very different. From the diagnostic point of view, celiac disease has clear diagnostic markers (serology, genetic susceptibility, and intestinal biopsy), while NCGS has no established diagnostic markers [20]. Moreover, in spite of some similar clinical symptoms, celiac disease is associated with more severe clinical presentation (short stature, malabsorption, autoimmune diseases, etc.) and significant mucosal immune changes compared to NCGS [21,22]. In fact, by definition, the diagnosis of NCGS can be established only after celiac disease is ruled out. Nonetheless, in cases with gluten intolerance with mild symptoms, we cannot predict who will develop full blown celiac disease and who will develop NCGS.

## 6. IBS and NCGS

The similar symptoms of NGCS and IBS may indicate that those two diseases overlap. Indeed, gluten protein, amylase-trypsin inhibitors (ATI) and fructan have been shown to trigger symptoms in both diseases, suggesting that they may share similar pathophysiology [15,23–25]. Indeed, treatment with a gluten-free diet (GFD) and a low fermentable oligosaccharides, disaccharides, monosaccharides, and polysaccharides (FOD-MAP) diet is helpful for IBS and NCGS [16,26,27]. It has also been suggested that IBS patients without minimal laboratory or histological support for celiac disease (lymphocytic duodenitis or positive HLA type) should not be on GFD, as it will not ameliorate their symptoms [10].

Although the etiology of both diseases is unknown, it was hypothesized that increased alteration in intestinal permeability (gut leakage) and changes in the intestinal microbiota were possibly the pathophysiological options for both diseases [25,28,29]. However, other studies showed that NCGS and IBS are not the same. The overall global incidence of NCGS and IBS are different (approximately 10% vs. up to 30%, respectively) [30], and clinically, NCGS typically does not fulfill the IBS clinical criteria (ROME 4). It is hypothesized that unlike NCGS, IBS is associated with various food intolerances while NCGS is associated with immunological changes (higher expression of CD63 compared to IBS) [31]. Moreover, although there are no established diagnostic markers for either disease, studies showed that in NCGS patients, increased rates of antigliadin antibody (IgG) (average rate- 50%) and a higher number of IEL (CD3 T cells) and eosinophils were noted in the duodenal and colonic mucosa compared to IBS patients [32,33]. The difference in prevalence rate between NCGS and IBS suggests that only a small number of IBS patients overlap with NCGS disease, which is currently defined as NCGS with IBS-like symptoms.

Overall, both NCGS and IBS are common in the general population and may coexist without sharing the same etiopathology. It is likely that both diseases are heterogeneous, with many possible contributing factors such as intestinal microbiota, intestinal permeability, and different degrees of gut mucosal inflammation [22]. Until we have better diagnostic markers for NCGS, NCGS with "IBS-like disease" will define the patients who have both diseases. At present, the data suggest that NCGS is a real disease that partially shares clinical symptoms with IBS.

Summary Points:

1. Nonceliac gluten sensitivity (NCGS) is a gluten-related disorder that is different than celiac disease;
2. NCGS has intestinal and extraintestinal symptoms that overlap with irritable bowel syndrome (IBS);
3. NCGS has not been associated with future malignancy;
4. NCGS has no specific clinical or laboratory markers; thus, the diagnosis was established clinically after celiac disease had been ruled out;
5. Although the clinical symptoms of IBS and NCGS are similar, NGCS is a real clinical phenomenon that awaits specific diagnostic markers.

**Conflicts of Interest:** The authors declare no conflict of interest.

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
