# Peer review of "Nonceliac Gluten Sensitivity—A Masquerading IBS or a Real Phenomenon?"

_gastrointestdisord, doi:10.3390/gidisord2020011_

Round 1

Reviewer 1 Report

Thi is a well-written and interesting commentary about the relationship between IBS and NCGS.

I have only minor comments.

  1. The authors should deepen the pathophysiologic role of wheat in non celiac gluten/wheat sensitivity. Wheat is only briefly cited in the introduction.

  2. GFD is an abbreviation not specified as gluten free diet inside the article.

  3. With respect to IBS and NCGS, the authors should specify that only a part of IBS patients ameliorates during a GFD compared to the totality of NCGS patients. This is one of the main clinical difference between these diseases.

Author Response

We appreciate and thank reviewer 1 for his comments. Enclosed please find point by point our response.

  1. Gluten protein is a part of the whole wheat protein. Unfortunately the exact culprit of the wheat/gluten protein has not been identified in NCGS disease. Accordingly, in the NASPGHAN diagnostic criteria established for NCGS (JPGN 2016) Hill I and his colleagues reviewed this topic and showed the overlap symptoms and differences among celiac disease, NCGS and wheat allergy. They clearly included wheat allergy in the differential diagnosis of gluten related diseases in human and suggested that wheat allergy should be excluded before NCGS could be established. We added a short highlighted sentence in the manuscript to clarify this point.
  2. GFD has been spelled out as requested.
  3. Although NCGS and IBS have shared a similar pathophysiological pathway and share similar therapies, not all IBS patients will improve on GFD alone (Am J Gastro. 2009). This point has been emphasized in the manuscript as per the reviewer comment.

Reviewer 2 Report

The comment is of high interest. NCGS needs to be further investigated and biomarkers must be determined in order to better identify and treat the patients.

This comment focuses on most important recent advances and it is well presented and designed.

Some little comments are included in the pdf document.

Reviewer 3 Report

  1. The authors affirm: “the rates of missed or overdiagnosis of NCGS were high”. This term "overdiagnosis" is widely used in the medical literature but in some scenarios, it does not reflect reality. In the case that the authors are mentioning it is not really an overdiagnosis but a true diagnostic error often with serious consequences on the state of nutrition and costs (Gluten-free diet is expensive).
  2. The authors should include other extra-intestinal (better extra-digestive) manifestations such as rhinitis, dermatitis, migraine and arthralgia. At least in the adult population they are very frequent and should be intentionally tested and sought in the history.
  3. The term "gluten intolerance" is discouraged by the Oslo Consensus (Ludvigsson JF, Leffler DA, Bai JC, Biagi F et al; The Oslo definitions for coeliac disease and related terms. Gut. 2013 Jan;62(1):43-52 ) as it is very confusing. “…we believe the term gluten intolerance is non- specific and carries inherent weaknesses and contradictions. Although gluten intolerance could be a consequence of poor digestion, it could also be the effect of some lectin-like properties of gluten or foods generated from gluten that cause GI upset. Another problem is that gluten intolerance may not truly reflect intolerance to gluten but to other wheat components.156 Because of these contradictions, we recommend that the term gluten intolerance should not be used and that gluten-related disorders be used instead of…” Definitely, this term, widely used in previous decades, even to formally define celiac disease, should not be used in a biomedical field.
  4. In their manuscript, the authors make the following statement: The overall global incidence of NCGS and IBS are different (approximately 10% vs. up to 30%, respectively). At least in the adult population and, according to the new criteria of Rome IV, this value is very oversized. If this statement is accepted, we would be transmitting a discouraging message for the general population. These data should be reviewed.
  5. The increased rates of anti-gliadin antibody (IgG) (average rate- 50%), observed in some studies, have not been sufficiently ratified in the literature and at least in our experience, they do not correspond to the reality of clinical practice.
  6. The importance of genetic study (HLA DQ2/DQ8) is not mentioned in the manuscript. It has been postulated that a negative result makes a consistent diagnosis of celiac disease highly unlikely. At this point, patients with symptoms suggestive of possible or probable NCGS and a negative result of HLA DQ2/DQ8 would not require a duodenal biopsy to exclude celiac disease (CD), which saves costs for the healthcare system. This is an important reflection on an entity whose symptoms overlap with both celiac disease and irritable bowel syndrome.
  7. A special consideration concerns the group of patients with compatible symptoms of either CD or NCGS who do not express serum autoantibodies, but have a positive genetic test for HLA DQ2 / DQ8 and exhibit mild forms of enteropathy in the duodenal mucosa (Marsh 1-Marsh 3a). Some authors recommend in these cases the use of advanced diagnostic techniques. A coeliac lymphogram, defined as an increase in CD3+ TCRγδ+ intraepithelial lymphocytes plus a decrease in CD3− intraepithelial lymphocytes, assessed by flow cytometry in duodenal biopsy samples is associated with a high level of diagnostic evidence in favour of CD. This is an important point to differentiate a NCGS of a true CD, especially in cases of lymphocytic enteritis (Marsh 1)-(Fernández-Bañares F, Crespo L, Núñez C, López-Palacios N, Tristán E, Vivas S, et al. Gamma delta+ intraepithelial lymphocytes and coeliac lymphogram in a diagnostic approach to coeliac disease in patients with seronegative villous atrophy. Aliment Pharmacol Ther. 2020 Feb 12. doi 10.1111/apt.15663.)
  8. The importance of some components of cereals such as FODMAPs and trypsin amylase inhibitors is named in the article as responsible for the symptoms in some patients with "NCGS". It would be important to highlight that a high proportion of patients present worsening of symptoms with placebo in the double-blind challenge tests (nocebo effect). This point highlights the importance of mistakenly attributing the nature of symptoms to gluten, when its origin is different (for example, abuse of fermentable sugars and / or intestinal bacterial overgrowth). At least in the adult population this is a well-proven fact.

Author Response

We appreciate reviewer #3 comments and would like to respond to his comments.

  1. the reviewer is "concern" about the term "over diagnosis of NCGS". We agreed with his comment and removed the term while left the "missed" diagnosis as the proper term. Overall, it is undeniable that due to lack of diagnostic criteria for NCGS, most of self declared pts with NCGS are missed diagnosed even under DBPC studies (ref. 7, 3).
  2. The reviewer correctly mentioned the many other symptoms of adult patients with NCGS. Indeed in the cited references (8, 9) many other symptoms were reported. In this paragraph ("Symptoms") we concentrated on the symptoms that may overlap NCGS with IBS and not with all the symptoms noted in adult with NCGS. Accordingly, we limited the list of the symptoms to that issue. The many other non intestinal symptoms reported in adults with NCGS could be reviewed in the cited references.
  3. The reviewer commented that in 2013, Oslo definition has abandoned the term "gluten intolerance" and adopted the new term "gluten related disorders". Indeed that nomeclature has been adopted by another committee (BMC Medicine 2012, 10:13). Nonetheless, other expert committee on this subject has not removed this term as reflected by a later publication from 2015 (non celiac gluten sensitivity, Gastroenterology 2015, 148: 1195). We believed that NCGS term should continue under the umbrella of "gluten related disorders" that included other diseases such as autoimmune dis.(celiac), allergic dis. (wheat allergy), and other. Moreover, NCGS terminology has been "adopted" by the PCPs as well as the public and for that reason should be left as is.
  4. The overall rate of IBS was cited from a publication in 2001 according to ROME 3 criteria (ref 32). Indeed when ROME 4 criteria was introduced in 2016, the epidemiology and global rates were changed. When ROME 3 criteria was compared to ROME 4 criteria, the rate of IBS was decreased. Overall the rate was reduced by half (J Gastro Hepatol 2017, 32: 2018). Those number has now introduced to the manuscript (line 102-104).
  5. We agree with the reviewer comments. Not all publications cited 50% increase of AGA-IgG in NCGS and for that reason we wrote "up to 50%". We have now modified the rates and added ref 8 to support the reviewer comment. (line 109-110).
  6. We do not understand the reviewer comment. We have clearly written that celiac disease has its own specific clinical or genetic markers while NCGS has no such markers (Line 84). This was also been published by the expert committee on NCGS report (ref. 4). Although negative genetic markers will exclude celiac disease, up to 20% of the population may be positive irrespective of NCGS disease. Accordingly, the discussion of genetic celiac markers to diagnose or exclude NCGS is not needed.
  7. We appreciate the comment of the reviewer. Nonetheless, the referred reference of the reviewer (Aliment Pharmacol Ther 2020) deals with possible celiac pts who had negative genetics with positive mild histology. This situation has nothing to do with NCGS disease. In that manuscript TCR-gamma-delta positive IEL receptors and celiac lymphogram were used to diagnosed celiac in pts with negative genetics. The authors concluded that these methods will help diagnosed celiac disease in pts with mild histology and negative genetics. The authors did not suggested or implied that those with negative results should be considered as NCGS. We do not think that this important publication has anything to do with NCGS and only assess the possible diagnosis of celiac in pts who have negative serology and mild histology. Accordingly, we do not feel that this ref. should be included in our review of NCGS disease.  
  8. We agree with the reviewer that nocebo effect is a big part of patients who are self diagnosed with NCGS. Indeed several paper (Placebo controlled) showed that FOD-MAP diet rather than gluten is responsible for their symptoms (ref. 17). Indeed we have written that IBS pts with negative minimal support markers for celiac disease should not be on GFD (ref 10, line 97-99). We have now added a sentence to support the comment of the reviewer (line 97).

Round 2

Reviewer 3 Report

I agree with the authors' comments and reply. Thank You..!